# Involvement of Multidrug Resistance Modulators in the Regulation of the Mitochondrial Permeability Transition Pore

**DOI:** 10.3390/membranes12090890

**Published:** 2022-09-16

**Authors:** Tatiana Fedotcheva, Nikolai Shimanovsky, Nadezhda Fedotcheva

**Affiliations:** 1Science Research Laboratory of Molecular Pharmacology, Medical Biological Faculty, Pirogov Russian National Research Medical University, Ministry of Health of the Russian Federation, Ostrovityanova St. 1, Moscow 117997, Russia; 2Institute of Theoretical and Experimental Biophysics, Russian Academy of Sciences, Institutskaya St. 3, Pushchino 142290, Russia

**Keywords:** multidrug resistance, mitochondrial permeability transition pore, quinidine, verapamil, avermectin

## Abstract

The permeability transition pore in mitochondria (MPTP) and the ATP-binding cassette transporters (АВС transporters) in cell membranes provide the efflux of low-molecular compounds across mitochondrial and cell membranes, respectively. The inhibition of ABC transporters, especially of those related to multi drug resistance (MDR) proteins, is an actively explored approach to enhance intracellular drug accumulation and increase thereby the efficiency of anticancer therapy. Although there is evidence showing the simultaneous effect of some inhibitors on both MDR-related proteins and mitochondrial functions, their influence on MPTP has not been previously studied. We examined the participation of verapamil and quinidine, classified now as the first generation of MDR modulators, and avermectin, which has recently been actively studied as an MDR inhibitor, in the regulation of the MPTP opening. In experiments on rat liver mitochondria, we found that quinidine lowered and verapamil increased the threshold concentrations of calcium ions required for MPTP opening, and that they both decreased the rate of calcium-induced swelling of mitochondria. These effects may be associated with the positive charge of the drugs and their aliphatic properties. Avermectin not only decreased the threshold concentration of calcium ions, but also by itself induced the opening of MPTP and the mitochondrial swelling inhibited by ADP and activated by carboxyatractyloside, the substrate and inhibitor of adenine nucleotide translocase (ANT), which suggests the involvement of ANT in the process. Thus, these data indicate an additional opportunity to evaluate the effectiveness of MDR modulators in the context of their influence on the mitochondrial-dependent apoptosis.

## 1. Introduction

The permeability transition pore in mitochondria (MPTP) and ATP binding cassettes transporters (АВС transporters) in cell membranes are the systems of non-specific membrane transport, which provide the efflux of low-molecular compounds across mitochondrial and plasma membranes, respectively. Both are of fundamental importance for cell survival in normal and pathological conditions. MPTP provides the flux of low-molecular-weight solutes across the generally impermeable inner mitochondrial membrane and stimulates apoptosis, and ABC transporters form a pathway for the translocation of different substrates across the plasma membrane and protect cells from the action of toxic metabolites and xenobiotics.

ABC transporters promote the efflux of various compounds, including toxic metabolites and therapeutic drugs applied in the treatment of many diseases. Removing drugs from cells, ABC transporters play a key role in the development of the resistance to anticancer drugs, a phenomenon known as multidrug resistance (MDR). P-gp and some other members of the ABC-transporter superfamily, related to MDR, involves transmembrane proteins that use the energy of ATP to withdraw substrates through a membrane. These transporters have two cytoplasmic nucleotide-binding domains (NBD), which hydrolyze ATP, and two transmembrane domains (TMD), which bind substrate molecules and form a pathway for their translocation across the membrane [1,2,3].

The inhibition of ABC transporters is an actively employed approach to increase the efficiency of anticancer therapy. Numerous compounds have demonstrated the ability to inhibit the transport activity of P-glycoprotein and to evoke intracellular drug accumulation. Among these are drugs such as the anti-arrhythmia agent verapamil, the immunosuppressive drug cyclosporin A, and the antimalarial anti-arrhythmia drug quinidine [1,4]. These inhibitors are now classified as the first generation of MDR modulators [1,5]. Quinidine not only inhibits Pgp but also blocks sodium and potassium channels and affects α-adrenergic receptors [6], verapamil inhibits voltage gated calcium channels [7], cyclosporin A (CsA) lowers the cyclophilin A-dependent signaling and transcription, induces immunosuppression, and is also well known as the effective inhibitor of the calcium-dependent MPTP opening [8,9].

The inhibitory effects of CsA is mediated by cyclophilin D, a regulator of Ca^2+^-induced MPTP opening. It is assumed that the formation of MPTP is mediated by several participants, including cyclophilin D, adenine nucleotide translocase (ANT), ATP synthase, a phosphate carrier, and the outer membrane voltage-dependent anion channel [8,9]. MPTP is involved in the regulation of both physiological and pathological cellular processes, with a low conductance associated with the redistribution of ions and small metabolites, which occurs in physiological conditions, and with a high conductance related to the efflux of large molecules, which influences mitochondrial functions and can lead to cell death [9]. As was supposed, both ATP synthase and adenine nucleotide translocase are required for the formation of a fully functional MPTP channel [8,9,10]. 

There are data showing the simultaneous effect of inhibitors on both MDR proteins and mitochondria. Thus, verapamil was found to attenuate oxidative stress and to prevent mitochondrial injury [11], and protect against the mitochondrial dysfunction in transient ischemia/reperfusion [12]. As was shown, some polyphenols in combination with verapamil induce a synergistic effect on P-gp [13] and can regulate the opening of MPTP by themselves [14]. Also, we found that the influence of some natural and synthetic steroids extends not only to the transport activity of MDR proteins, but also to the regulation of MPTP [15,16,17]. It was revealed that adenine nucleotide translocase (ANT) is a mitochondrial target for some compounds of this group of steroids. Thus, these data indicate that the same modulators can affect both transport systems.

In this work, we studied the influence of MDR inhibitors on the opening of MPTP. In addition to verapamil and quinidine, we tested the effect of macrocyclic lactone avermectin, which has recently been actively studied as an MDR inhibitor [18,19]. Also, there is evidence on the influence of avermectin on mitochondrial functions, namely, oxidative phosphorylation and the membrane potential maintenance [20]. The chemical structures of the tested compounds are shown in the Figure 1. We examined the influence of these three MDR inhibitors on Ca^2+^-induced MPTP opening and the possible involvement of ANT in their effects.

## 2. Materials and Methods

### 2.1. Reagents and Chemicals

All reagents were from the Sigma–Aldrich Corporation (St. Louis, MO, USA). Avermectin (Avermectin B1) was from FarmBioMed, Russia, Moscow.

### 2.2. Preparation of Rat Liver Mitochondria

Mitochondria were isolated from adult Wistar male rats. The study was conducted in accordance with the ethical principles formulated in the Helsinki Declaration on the care and use of laboratory animals. Manipulations were carried out by the certified staff of the Animal Department of the Institute of Theoretical and Experimental Biophysics (Russian Academy of Sciences and approved by the Commission on Biomedical Ethics of ITEB RAS (N2/2022, 03/05/2022). During the study, the animals were kept in wire-mesh cages at room temperature with a light/dark cycle of 12 h. Mitochondria from the liver of anesthetized animals were isolated using the standard method. The liver was rapidly removed and homogenized in an ice-cold isolation buffer containing 300 mM sucrose, 1 mM EGTA, and 10 mM HEPES–Tris (pH 7.4). The homogenate was centrifuged at 600× *g* for 7 min at 4 °C, and the supernatant fraction was then centrifuged at 9000× *g* for 10 min to obtain mitochondria. Mitochondria were washed twice in the above medium without EGTA. The final mitochondrial pellet was suspended in the washing medium to yield 60 mg protein/mL and kept on ice for analysis. Mitochondria with a respiratory control ratio of about 4, as measured by the polarographic method with succinate as substrate [17], were used throughout all experiments.

### 2.3. Determination of the Ca^2+^-Induced MPTP Opening

The opening of the MPTP was registered as a steep rise in calcium ions in the incubation medium and dissipation of the mitochondrial membrane potential [14,17]. MPTP opening was induced by the sequential loading of the incubation medium with CaCl_2_. The difference in the electric potential on the inner mitochondrial membrane was measured from the redistribution of lipophilic cation tetraphenylphosphonium (TPP^+^) between incubation medium and mitochondria. The concentration of TPP^+^ in the incubation medium was recorded by a TPP^+^ selective electrode (Nico, Moscow, Russia). Changes in the calcium ion concentration in the incubation medium were recorded by a Ca^2+^-selective electrode (Nico, Moscow, Russia). The concentration of TPP^+^ and Ca^2+^ was registered simultaneously in an open chamber of volume 1 ml containing 1.0–1.2 mg mitochondrial protein under continuous stirring. MPTP opening was induced by the sequential loading of the incubation medium with 25 μM Ca^2+^ (CaCl_2_). The mitochondrial calcium retention capacity (CRC) was determined as the total concentration of added Ca^2+^ required for pore opening. The incubation medium contained 125 mM KCl, 15 mM HEPES, 1.5 mM phosphate, 5 mM succinate, pH 7.25.

### 2.4. Determination of Swelling of Mitochondria

The swelling of mitochondria was measured at a wavelength of 540 nm using an Ocean Optics USB4000 spectrophotometer (Ocean Optic, Dunedin, FL, USA). Swelling was assessed by measuring the changes in optical density during incubation. Mitochondria at a concentration of mitochondrial protein of 0.3–0.4 mg/ml were incubated in the same buffer (125 mM KCl, 15 mM HEPES, 1.5 mM phosphate, 5 mM succinate).

### 2.5. Statistical Analysis

The data given represent the means ± standard error of means (SEM) from five to seven experiments or are the typical traces of three to five identical experiments with the use of different mitochondrial preparations. The statistical significance was estimated by the Student’s *t*-test with *p* < 0.05 as the criterion of significance.

## 3. Results

### 3.1. The Effect of Drugs Was Assessed by Changing the Threshold Concentrations of Calcium Ions Required to Open MPTP

Figure 1a shows the changes in the membrane potential during successive additions of CaCl_2_ in the presence of quinidine in the concentration range from 25 to 100 µM. At drug concentrations of 25 and 50 µM, the decrease in the threshold concentrations of calcium ions was insignificant. At a concentration of 100 μM, quinidine decreased the calcium retention capacity by 25%. Besides, the addition of quinidine decreased the membrane potential and declined the magnitude of the response to the addition of calcium ions. Verapamil had a similar effect on the membrane potential, but in contrast to quinidine, it increased the threshold concentration of calcium ions that opens MPTP, i.e., increased the calcium retention capacity of mitochondria (Figure 1b). With an increase in the concentration from 25 to 100 μM, verapamil increased the calcium retention capacity from 20 to 50%. 

The measurements with a calcium selective electrode showed that both drugs induce a rise similar to that induced by calcium supplementation (Figure 1c). This rise was not removed in the presence of EGTA (Figure 1c insert) and is probably due to the positive charges and the amphiphilic properties inherent in quinoline- and phenylalkylamine-based drugs, quinidine and verapamil, respectively. In these tests, the influence of quinidine and Verapamil was similar to their effects on the membrane potential as an indicator of MPTP opening in response to successive additions of calcium ions. At concentrations of 100 μM, quinidine activated and verapamil inhibited the Ca^2+^-induced opening of MPTP (Figure 1d). 

The above side effects are absent in the spectral measurements of mitochondrial swelling as an indicator of pore opening, which allows the use of higher concentrations of drugs. As shown in Figure 2a, verapamil had no effect on mitochondrial swelling even at concentrations as high as 200–250 µM (or 500–600 µM per mg of mitochondrial protein), while quinidine induced low-amplitude swelling at concentrations up to 200 µM. Both drugs lowered the rate of high-amplitude swelling induced by calcium ions. 

The influence of quinidine was weaker compared to verapamil (Figure 2b,c). The inhibition of the swelling rate did not exceed 15–20% at concentrations of 100–200 μM quinidine and reached 35–40% at the same concentrations of verapamil (Figure 2d).

### 3.2. Influence of Avermectin on the MPTP Opening Induced by Calcium Ions

In contrast to quinidine and verapamil, avermectin had a pronounced specific influence on MPTP even at much lower concentrations. As shown in Figure 3, Avm at concentrations of 25 and 50 µM reduced the threshold calcium ion concentrations by 20% and 50%, respectively. At a concentration of 100 μM, avermectin activated the opening of the pore immediately after the first addition of 25 μM calcium ions (Figure 3b). The addition of MPTP inhibitors CsA and ADP partially eliminated the effect of avermectin. CsA prevented the opening of MPTP, and ADP increased the threshold calcium concentration several times (Figure 3b). Also, it can be seen (Figure 3a,b) that the addition of avermectin caused a concentration-dependent decrease in the membrane potential, which was then stabilized at a lower level and did not restore in the presence of CsA and ADP. The absence of their influence indicates that the decrease in the membrane potential is not associated with the MPTP opening, but rather is an additional property of avermectin, which also had been noted earlier [20].

The protective effect of CsA and ADP indicates the participation of ANT in the action of avermectin on MPTP, which is confirmed by the following experiments using the selective ANT inhibitor carboxyatractyloside (Catr). Catr almost doubled the effect of avermectin on MPTP, which manifested itself as a drop in the membrane potential (Figure 3c) and the release of calcium ions from mitochondria (Figure 3d) already at low concentrations of added calcium. Thus, in these tests, avermectin in the concentration range of 25–100 μM activated the opening of calcium-induced MPTP by decreasing the calcium retention capacity (Figure 3e); the effect was suppressed by CsA and ADP and enhanced by Catr (Figure 3f). 

Then, we examined the effect of avermectin on the mitochondrial swelling as one of the main indicators of MPTP opening. It turned out that avermectin is able to induce the swelling by itself in the absence of added calcium. As shown in Figure 4a, avermectin induced low-amplitude swelling at a concentration as low as 10 µM. The amplitude and rate of the swelling increased many times with an increase in avermectin concentration to 20 and 50 μM (Figure 4b). It is important that CsA completely prevented, while Catr, on the contrary, strongly activated the avermectin-induced swelling (Figure 4c). ADP had a protective effect, increasing the avermectin concentration required to induce swelling (Figure 4d). These data show that the avermectin-induced swelling is associated with the MPTP opening and the involvement of ANT in this process.

## 4. Discussion

The study has revealed that the action of MDR inhibitors extends to the regulation of MPTP opening. In experiments on rat liver mitochondria, quinidine and avermectin lowered the threshold concentrations of calcium ions required to open MPTP. Avermectin, in addition to this effect, induced the MPTP opening by itself. In contrast, verapamil increased the threshold concentrations of calcium ions. All three inhibitors showed a direct effect on mitochondria, as evidenced by both the calcium ion accumulation and the swelling of mitochondria. Thus, these data indicate an additional opportunity to evaluate the effectiveness of MDR inhibitors. As known, some of them act directly on the ATPase activity or the NBD domain of Pgp, while others act indirectly, by influencing the expression of these proteins. The first are quinidine and verapamil, the latter include avermectin, which blocks proliferative signaling pathways [21].

Previously, the influence of quinidine and verapamil, as anti-arrhythmic drugs, on mitochondrial functions has been studied. It was found that quinidine partially blocks the mitochondrial voltage-dependent anion channel [22], inhibits the ATP-sensitive potassium channel [23], decreases the respiratory control index [24] and inhibits myocardial mitochondrial ATPase [25]. Verapamil demonstrated the protective action against mitochondrial dysfunction and apoptosis in conditions of ischemia/reperfusion [12]. Also, verapamil attenuated the pacing-dependent drop in endogenous respiration and ATP levels in cardiomyocytes [26]. As for avermectin and its analogs, it has been shown to inhibit mitochondrial respiration by decreasing the activity of respiratory complex I and to induce mitochondria-dependent apoptosis [21,27].

Our study revealed for the first time the involvement of these inhibitors in the regulation of MPTP opening. The most active of them was Avm, which induced the opening of MPTP also in the absence of added calcium. Moreover, it acted at concentrations comparable to those modulating MDR, which range from 5 to 40 µM [20,28].

The same refers to verapamil and quinidine, for which the effective concentrations on cells varied, according to different data, from 5 to 500 µM. Thus, quinidine at concentrations about 20 µM potentiated the toxicity of doxorubicin in cancer cell overexpressing a multidrug resistance associated protein [29]. In cultured rabbit corneal epithelial cells, the accumulation of the drug increased up to 19% or 36% at 500 µM concentrations of verapamil or quinidine, respectively [30]. In our experiments, the modulation of MPTP opening was observed at concentrations of 50–100 μM for verapamil and quinidine, and of 10 μM for avermectin, which is consistent with their concentrations as MDR inhibitors.

In the tests of the membrane potential and calcium retention capacity, verapamil and quinidine manifested properties as membrane-active compounds. Both drugs affected the sensitivity of measurements to calcium ions and TPP, which may be due to the positive charge of the molecules and their aliphatic properties. As was earlier noted, these characteristics can influence the binding and partitioning of verapamil with membrane lipids due to adsorption and electrostatic and hydrophobic interactions, thereby contributing to the functional activity of the drug [31,32]. It was also suggested that the positive-charged state of quinidine is partly responsible for the inhibition of the calcium release from the cardiac sarcoplasmic reticulum [33]. The above properties may also contribute to the modulation of MPTP by verapamil and quinidine observed in our experiments. Since both drugs are ion channel inhibitors, they could affect the uptake of calcium ions, which is carried out by the potential-dependent calcium uniporter (MCU). They did not induce pronounced changes in the calcium uptake, but changed the threshold calcium concentrations required for the pore opening. The decline in the calcium retention capacity may be partly associated with the membrane potential decrease provoked by the drugs. Besides, they both decreased the rate of calcium-induced swelling of mitochondria, which can be explained by the aliphatic properties of the compounds, i.e., the incorporation into membranes and the interaction with lipids.

In contrast, avermectin not only decreased the threshold concentrations of calcium ions, but also itself induced the opening of MPTP by itself, which was especially pronounced in the induction of mitochondrial swelling. Of greatest interest are the data showing the involvement of ANT in this process, which is indicated by the partial inhibition of the avermectin-induced swelling by low concentrations of ADP and the activation of the swelling by Catr. ADP and Catr are typical regulators, an inhibitor and an activator, respectively, of the opening of cyclosporin A-sensitive MPTP. These results indicate the participation of ANT in the opening of MPTP induced by avermectin. It is known that ATR locks the ANT in the “c” conformation (nucleotide binding site facing the cytosol) [8]. As was established, ADP fixes ANT in the m-state and inhibits pore opening, while the c-state of ANT is one of the necessary conditions for MPTP opening [34]. It is possible that avermectin activates the pore opening in a similar manner, inducing the swelling with a different amplitude, depending on the concentration. The involvement of ANT in the effect of the avermectin, which acted like Catr and oligomycin on the mitochondrial oxidative phosphorylation, was also shown to be a potential mechanism of hepatotoxicity [35]. Avermectin also enhanced apoptosis, up-regulating the expression of mitochondrial pro-apoptotic proteins [20]. 

As was previously noted, conformational changes of ANT can be associated with the cisplatin resistance since carboxyatractyloside improved the chemosensitivity of cancer cells [36]. It was suggested that ANT could serve as a novel therapeutic target for overcoming the cisplatin resistance. Our results support the assumption concerning the involvement of ANT in the action of inhibitors aimed at overcoming the drug resistance. It can be assumed that these two processes are independent of each other but are regulated by common modulators, in particular, pyridine nucleotides. According to current models, the binding of ATP promotes NBD dimerization, resulting in external accessibility of the drug-binding site (closed NBD conformation), and ATP hydrolysis leads to the dissociation of NBDs with the subsequent return of the accessibility of the binding site to the cytoplasmic side (open NBD conformation) [37] ATP hydrolysis and subsequent Pi or ADP release are required for resetting the transporter back to the inward-facing conformation to accept another substrate molecule [1,38].

Thus, the models describing the steps of the transport cycle include ATP binding and hydrolysis, ADP and inorganic phosphate (Pi) release and nucleotide dependent structural changes. These steps are linked to a conformational switch of the TMDs, which interchange between the high-affinity inward-facing state and the low-affinity outward-facing state [39]. This regulation of the transporter is similar in some characteristic features to the regulation of ANT conformational states in MPTP opening and blocking.

As our data show, the action of MDR inhibitors extends to the regulation of MPTP.

Of interest would be a study of next-generation inhibitors, the more so that some of them are the modified analogs of CsA, quinoline, and verapamil [40]. Such a study would provide an additional opportunity to evaluate the effectiveness of MDR modulators in the context of their simultaneous influence on the mitochondrial-dependent apoptosis. A relevant problem is how the blockage or the activation of MPTP affect the accumulation of drugs and MDR as a whole. As is known, mitochondria are involved in the redistribution of drugs in the cell, including the cases with the participation of ABC transporters. Thus, it was found that the mitochondria of doxorubicin-resistant breast cancer cells contain the multidrug resistance protein MRP1 in the inner membrane, which limits the mitochondrial accumulation of doxorubicin [41]. Moreover, mitochondria have their own class of ABC transporters that export various compounds to the cytosol. These include, for example, mitochondrial ABCB7-type exporters involved in the maturation of cytosolic iron-sulfur proteins via the export of [2Fe-2S] clusters [42], protein ABCB10 exporting heme precursor required for normal heme production [43] and ABCB8, an iron exporter, involved in iron export from mitochondria to cytosol [44]. In this context, of great interest are also the data showing the almost total dependence of ABC transporters on ATP produced by mitochondria [44,45], and relevant importance of mitochondrial dysfunction in reducing MDR [46,47].

## Data Availability

The data sets generated during the current study are available from the corresponding author on reasonable request.

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
