# Peer review of "Involvement of Multidrug Resistance Modulators in the Regulation of the Mitochondrial Permeability Transition Pore"

_membranes, 2022, doi:10.3390/membranes12090890_

Round 1

Reviewer 1 Report

See attached file

Author Response

Replay 1

We thank the reviewer for the interesting comments and remarks to our work. The reviewer noted the aspects that we also took into account during our experiments. Thus, we expected to see the inhibition of the MCU-mediated transport of calcium ions by these drugs, but they did not cause any pronounced changes in the calcium uptake. However, they induced significant modifications in such parameter as the calcium retention capacity, changing the threshold calcium concentrations required for the pore opening. This permeability transition pore strictly requires matrix Ca2+ and is inhibited by CsA, which is “the gold-standard inhibitor” [ref. 8] of the calcium-induced MPTP opening. If this pore was not inhibited by CsA, we would use other inhibitors such as phospholipase inhibitors, lipid radical scavengers, thiol antioxidants, etc. We agree that direct interactions of drugs with MPTP participants have not been established, but their involvement in the regulation of MPTP opening was found. We plan to continue these studies with new generations of MDR inhibitors and not only on isolated mitochondria, but also on cells.

We took into account all the comments and made appropriate additions and corrections to the manuscript.

Regarding MCU we have added the following paragraph to the Discussion section:

Since both drugs are ion channel inhibitors, they could affect the uptake of calcium ions, which is carried out by the potential-dependent calcium uniporter (MCU). They did not induce pronounced changes in the calcium uptake, but changed the threshold calcium concentrations required for the pore opening. The decline in the calcium retention capacity may be partly associated with the membrane potential decrease provoked by the drugs.

In the Introduction section, we have changed the phrase about CsA:

CsA lowers the cyclophilin A-dependent signaling and transcription, induces immunosuppression, and is also well known as the effective inhibitor of the calcium-dependent MPTP opening [8, 9].

We have added the following paragraph related to mitochondrial ABC transporters to the Discussion section:

….A relevant problem is how the blockage or the activation of MPTP affects the accumulation of drugs and MDR as a whole. As is known, mitochondria are involved in the redistribution of drugs in the cell, including the cases with the participation of ABC transporters. Thus, it was found that the mitochondria of doxorubicin-resistant breast cancer cells contain the multidrug resistance protein MRP1 in the inner membrane, which limits the mitochondrial accumulation of doxorubicin [41 ]. Moreover, mitochondria have their own class of ABC transporters that export various compounds to the cytosol. These include, for example, mitochondrial ABCB7-type exporters involved in the maturation of cytosolic iron-sulfur proteins via the export of [2Fe-2S] clusters [42 ], protein ABCB10 exporting heme precursor required for normal heme production [43] and ABCB8, an iron exporter, involved in iron export from mitochondria to cytosol [44]. In this context, of great interest are also the data showing the almost total dependence of ABC transporters on ATP produced by mitochondria [44, 45 ], and relevant importance of mitochondrial dysfunction in reducing MDR [46, 47].

  1. Dartier, J.; Lemaitre, E.; Chourpa, I.; Goupille, C.; Servais, S.; Chevalier, S.; Mahéo, K.; Dumas, JF. ATP-dependent activity and mitochondrial localization of drug efflux pumps in doxorubicin-resistant breast cancer cells. Biochim Biophys Acta Gen Subj. 2017, 1861(5 Pt A):1075-1084. doi: 10.1016/j.bbagen.2017.02.019.
  2. Li, P.; Hendricks, A.L.; Wang, Y.; Villones, R.L.E.; Lindkvist-Petersson, K.; Meloni, G.; Cowan, J.A.; Wang, K. Structures of Atm1 provide insight into [2Fe-2S] cluster export from mitochondria. Gourdon P. Nat Commun. 2022, 13(1):4339. doi: 10.1038/s41467-022-32006-8.
  3. Martinez M, Fendley GA, Saxberg AD, Zoghbi ME. Stimulation of the human mitochondrial transporter ABCB10 by zinc-mesoporphrin. PLoS One. 2020 Nov 30;15(11):e0238754. doi: 10.1371/journal.pone.0238754. eCollection 2020.
  4. Kumar, V.; Santhosh Kumar, T.R.; Kartha, C.C. Mitochondrial membrane transporters and metabolic switch in heart failure. Heart Fail Rev. 2019, 24(2):255-267. doi: 10.1007/s10741-018-9756-2.
  5. Giddings, E.L.; Champagne, D.P.; Wu, M.H.; Laffin, J.M.; Thornton, T.M.; Valenca-Pereira, F.; Culp-Hill, R.; Fortner, K.A.; Romero, N.; East, J.; Cao, P.; Arias-Pulido, H.; Sidhu, K.S.; Silverstrim, B.; Kam, Y.; Kelley, S.; Pereira, M.; Bates, S.E.; Bunn, J.Y.; Fiering, S.N.; Matthews, D.E.; Robey, R.W.; Stich, D.; D'Alessandro, A.; Rincon, M. Mitochondrial ATP fuels ABC transporter-mediated drug efflux in cancer chemoresistance. Nat Commun. 2021, 12(1):2804. doi: 10.1038/s41467-021-23071-6.
  6. Lee, A.C.K.; Lau, P.M., Kwan, Y.W.; Kong, S.K. Mitochondrial Fuel Dependence on Glutamine Drives Chemo-Resistance in the Cancer Stem Cells of Hepatocellular Carcinoma. Int J Mol Sci. 2021, 22(7):3315. doi: 0.3390/ijms22073315.
  7. Bokil, A.; Sancho, P. Mitochondrial determinants of chemoresistance. Cancer Drug Resist. 2019, 2(3):634-646. doi: 10.20517/cdr.2019.46.

Minor points:

Key references to the methods used are missing.

We have added references 14, 17 to the methods

What is the concentration of Ca added in experiments of Figure 2?

We indicated the concentration of calcium (25 μM) in the legend to the figure

Reviewer 2 Report

In the present work, the authors investigated the effect of the multidrug resistance modulators verapamil, quinidine, and avermectin B1 on the induction of the mitochondrial permeability transition. The authors applied the classical methods of studying this phenomenon and obtained some new results. However, the work needs significant revision before publication.

Comments:

The authors write that cyclosporin A is well known as the most effective inhibitor of MPTP. This is not entirely true, since many other inhibitors, including cyclosporins, have similar efficacy. Rather, it should be said that it is the most well-known MPT pore inhibitor.

The authors write that Mitochondria from the liver of anesthetized animals were isolated using the standard method. What was used for anesthesia?

Mitochondria with a respiratory control ratio of about 4 were used throughout all experiments. What is the substrate of respiration and what medium is used to assess respiration? This needs to be clarified.

The common abbreviation for cyclosporine A is CsA, as there are several types of cyclosporins.

The authors are recommended to give the structures of the studied compounds.

There are no asterisks in figure 4b.

The authors discuss rather weakly the involvement of ATP synthase in the induction of the MPT pore under their conditions. Recent work shows that ANT is involved in the induction of the low-conductivity state of the pore, while ATP synthase is involved in the induction of the high-conductivity state. The results of the authors may indicate the involvement of ANT in the processes under study, but do not exclude the participation of ATP synthase.

What is meant by the swelling amplitude in Fig. 4 (how long after swelling initiation was this parameter measured)? Units of measurement are not given. This also applies to the swelling rate. In general, these parameters should also be recalculated for the amount of protein.

Calcium retention capacity is given in relative units, however, at least for control, it is necessary to provide absolute data in the figure caption.

What causes the drop in membrane potential and the release of calcium induced by quinidine and verapamil in Fig. 1? Did CsA influence this fall? It is advisable to provide these data.

When assessing mitochondrial swelling, the authors used much less mitochondrial protein than in the case of assessing potential and calcium capacity. However, they significantly increased the concentration of the studied agents (quinidine and verapamil). This looks incorrect, the authors should equate the concentration of quinidine and verapamil to mitochondrial protein in these experiments.

On fig. 2, the authors evaluated the effect of quinidine and verapamil on mitochondrial swelling in the absence and presence of calcium. It can be seen that in the important control experiment (panel A), the initial optical density of the mitochondrial suspension is much higher than in panels B and C. This indicates a different level of mitochondrial protein or the state of organelles and the quality of their isolation. It is necessary to perform these experiments on the same population of mitochondria, under the same conditions and protein level. There is reason to believe that quinidine and verapamil will have more pronounced effects under these conditions, which needs to be explained.

One last note for the future. It is not recommended to use high concentrations of CsA in experiments (like 2 μM in this study), since at high concentrations this agent can affect the state of the lipid bilayer of mitochondrial membranes.   

Author Response

Comments:

We thank the reviewer for the useful remarks to our work. We have made appropriate corrections and additions to the manuscript.

The authors write that cyclosporin A is well known as the most effective inhibitor of MPTP. This is not entirely true, since many other inhibitors, including cyclosporins, have similar efficacy. Rather, it should be said that it is the most well-known MPT pore inhibitor.

We have clarified this phrase by emphasizing that inhibition of the Ca-induced pore is implied:

Cyclosporin A (CsA) lowers the cyclophilin A-dependent signaling and transcription, induces immunosuppression, and is also well known as the effective inhibitor of the calcium-dependent MPTP opening [8, 9].

The authors write that Mitochondria from the liver of anesthetized animals were isolated using the standard method. What was used for anesthesia?

 CO2 was used for anesthesia

Mitochondria with a respiratory control ratio of about 4 were used throughout all experiments. What is the substrate of respiration and what medium is used to assess respiration? This needs to be clarified.

We have done an addition:

Mitochondria with a respiratory control ratio of about 4, as measured by the polarographic method with succinate as substrate [17], were used throughout all experiments.

The common abbreviation for cyclosporine A is CsA, as there are several types of cyclosporins.

We have corrected this abbreviation throughout the text.

The authors are recommended to give the structures of the studied compounds.

We have added the chemical structures to the Introduction section.

There are no asterisks in figure 4b.

Figure 4 shows the dependence of the rate and amplitude of swelling on the concentration of the drug at a zero value of these parameters in the control, i.e. in the absence of swelling.

The authors discuss rather weakly the involvement of ATP synthase in the induction of the MPT pore under their conditions. Recent work shows that ANT is involved in the induction of the low-conductivity state of the pore, while ATP synthase is involved in the induction of the high-conductivity state. The results of the authors may indicate the involvement of ANT in the processes under study, but do not exclude the participation of ATP synthase.

The possible involvement of ANT follows from our data, but the contribution of ATPase was not evaluated in our experiments. We plan to explore this aspect with more suitable for this purpose methods.

What is meant by the swelling amplitude in Fig. 4 (how long after swelling initiation was this parameter measured)? Units of measurement are not given. This also applies to the swelling rate. In general, these parameters should also be recalculated for the amount of protein.

We added to the legend the recalculated values for the amount of protein:

Amplitude values are given in units of change in optical density for 200 s or upon reaching the stop of swelling. Swelling rates are given in units of change in optical density per minute.

Calcium retention capacity is given in relative units, however, at least for control, it is necessary to provide absolute data in the figure caption.

Although the control values could vary, especially during a long experiment, the influence of the drugs was observed in all cases. On freshly isolated mitochondria, this parameter was equal to an average of 110+12 μM at a mitochondrial protein concentration of 1 mg in the sample. We added the absolute value the calcium retention capacity for the control, indicated as 100%, to the legend:

The value of control calcium retention capacity (100%) is equal 110+12 μM at a concentration of mitochondrial protein of 1 mg.

What causes the drop in membrane potential and the release of calcium induced by quinidine and verapamil in Fig. 1? Did CsA influence this fall? It is advisable to provide these data.

At this stage of research, we believe that the decrease in the membrane potential can be explained by the amphiphilic properties and the positive charge of these drugs. We plan to clarify this aspect using more appropriate methods in future studies. We have added to Discussion:

Since both drugs are ion channel inhibitors, they could affect the uptake of calcium ions, which is carried out by the potential-dependent calcium uniporter (MCU). They did not induce pronounced changes in the calcium uptake, but changed the threshold calcium concentrations required for the pore opening. The decline in the calcium retention capacity may be partly associated with the membrane potential decrease provoked by the drugs.

When assessing mitochondrial swelling, the authors used much less mitochondrial protein than in the case of assessing potential and calcium capacity. However, they significantly increased the concentration of the studied agents (quinidine and verapamil). This looks incorrect, the authors should equate the concentration of quinidine and verapamil to mitochondrial protein in these experiments.

We agree that recalculation per mg of protein will raise the actual concentrations of supplements by about 2 times. We have completed the phrase about concentrations in Result section as follows:

As shown in Figure 2a, verapamil had no effect on mitochondrial swelling even at concentrations as high as 200-250 µM (or 500-600 µM per mg of mitochondrial protein).

On fig. 2, the authors evaluated the effect of quinidine and verapamil on mitochondrial swelling in the absence and presence of calcium. It can be seen that in the important control experiment (panel A), the initial optical density of the mitochondrial suspension is much higher than in panels B and C. This indicates a different level of mitochondrial protein or the state of organelles and the quality of their isolation. It is necessary to perform these experiments on the same population of mitochondria, under the same conditions and protein level. There is reason to believe that quinidine and verapamil will have more pronounced effects under these conditions, which needs to be explained.

This method is so sensitive that even small changes in the concentration of mitochondrial protein can increase the optical density. These changes lie in the range of 10%, and practically do not influence on the magnitude of the effects, especially at such high concentrations of drugs. We will take this remark into account in our studies.

One last note for the future. It is not recommended to use high concentrations of CsA in experiments (like 2 μM in this study), since at high concentrations this agent can affect the state of the lipid bilayer of mitochondrial membranes. 

Also, we will take this recommendation into account in our studies.

Round 2

Reviewer 2 Report

The authors have adequately addressed my questions and changed the text.